

# A novel technique including GPS radio occultation for detecting and monitoring volcanic clouds

Riccardo Biondi[1], Andrea Steiner[1], Gottfried Kirchengast[1,2], Hugues Brenot[3], Therese Rieckh[1]

[1]Wegener Center for Climate and Global Change (WEGC), University of Graz, Graz, Austria

[2]Institute for Geophysics, Astrophysics, and Meteorology/Institute of Physics, University of Graz, Graz, Austria

[3]Belgian Institute for Space Aeronomy (BIRA-IASB), Brussels, Belgium

Correspondence to: R. Biondi (riccardo@biondiriccardo.it)

## Abstract

The volcanic cloud top altitude and the atmospheric thermal structure after volcanic eruptions are studied using Global Positioning System (GPS) Radio Occultation (RO) profiles co-located with independent radiometric measurements of ash and $SO_2$ clouds. We use the GPS RO data to detect volcanic clouds and to analyze their impact on climate in terms of temperature changes. We selected about 1300 GPS RO profiles co-located with two representative eruptions (Puyehue 2011, Nabro 2011) and found that an anomaly technique recently developed for detecting cloud tops of convective systems can also be applied to volcanic clouds. Analyzing the atmospheric thermal structure after the eruptions, we found clear cooling signatures of volcanic cloud tops in the upper troposphere for the Puyehue case. The impact of Nabro lasted for several months, suggesting that the cloud reached the stratosphere, where a significant warming occurred. The results are encouraging for future routine use of RO data for monitoring volcanic clouds.

## 1. Introduction



Explosive volcanic eruptions produce large ash clouds and inject huge amounts of gas, aerosol,
and ash into the troposphere, which can even reach into the stratosphere (Bourassa et al., 2012,
2013; Fromm et al., 2013, 2014). Major volcanic eruptions can cause short-term climate change
(Robock, 2013) if sulfur dioxide ($SO_2$) is injected into the stratosphere, forming sulfate aerosols
with a long residence time (about 1 to 3 years). The effect is a global warming of the stratosphere
and a cooling of the troposphere as was observed for the Mount Pinatubo eruption (Robock,
2000). The impacts largely depend on the total mass erupted, the altitude reached by the ash and
$SO_2$ clouds, the location of the volcano, and the extent of the dispersion due to atmospheric
circulation. Under favorable atmospheric conditions volcanic ash clouds can spread over
thousands of kilometers in just a few hours.
Ash clouds are a threat for aviation transport (Prata, 2008), since they can damage the aircraft
engines even at large distances from the eruption. In 2010, the Eyjafjöll eruption in Iceland
(Stohl et al., 2011) generated the largest air traffic shutdown since the Second World War with
an estimated loss of about 3 billion dollars for the airline industry and with major effects on
social and economic activities. Research attention focused on the improvement of detection and
monitoring of volcanic ash clouds, which had already been advocated by Tupper et al. (2004).
The ESA-EUMETSAT workshop on "Monitoring volcanic ash from space" (Zehner, 2010)
provided a list of recommendations stating that "*Studies should be made of potential new*
*satellites and instruments dedicated to monitoring volcanic ash plumes and eruptions*" and
highlighting the difficulty to monitor such events with the current knowledge.
Observing the density of the ash cloud is one of the major challenges, since values larger than
2 mg/m$^3$ are considered dangerous for aircraft engines. This parameter can only be detected by
flying into the cloud with all related risks. The ejected mass of the eruption is fundamentally
related to the maximum height reached by a volcanic plume (Settle, 1978). This volcanic cloud
top altitude can be detected with different techniques (ground based, *in situ*, satellite), but
typically with quite low accuracy.
Knowledge of the cloud top altitude is essential, however, to provide information on ash-free
altitude regions for air traffic and on potential overshooting and spread of $SO_2$ into the
stratosphere, which impacts climate. The discrimination of ash clouds from other types of clouds
is challenging, wherefore Tupper et al. (2004) state "*a reliable detection system cannot be*





*dependent on the meteorological conditions and it is necessary to have a weather independent*

*warning capacity*". Along these lines the potential of the relatively new satellite technique of

radio occultation (RO) based on Global Positioning System (GPS) signals, or more generally

Global Navigation Satellite System (GNSS) signals, comes into play (Biondi et al., 2012, 2013).

In this study we provide an assessment of the potential capacity of the RO technique for volcanic

cloud detection and monitoring. Section 2 provides an overview of the available observing

techniques and introduces the potentially unique role of RO data. Section 3 then summarizes the

data sets used and section 4 the study cases (three example eruptions) and methods.

Subsequently we discuss the results in section 5 and draw conclusions in section 6.

## 2. Volcanic Cloud Observing Techniques

Volcanic ash clouds are currently monitored by the International Airways Volcano Watch

(IAVW) using a combination of ground-based sensors, satellite sensors, and aircraft

measurements, but each of these methods has some temporal, spatial or technological limitation.

According to the International Union of Geodesy and Geophysics (IUGG) only about 50% of the

World's volcanoes that currently threaten air operations have any sort of ground-based

monitoring (IUGG, 2010). The greatest danger for the air traffic is the time just after the eruption

when no warnings are available, models are not reliable, and atmospheric observations are

sporadic. The vertical resolution of most satellite data is very coarse for monitoring such kind of

phenomena and thus there is an urgent need to gather information on the vertical structure of

evolving volcanic clouds (Zehner, 2010).

Geostationary satellite data (e.g., the Spinning Enhanced Visible and InfraRed Imager - SEVIRI)

and polar satellite data (e.g., the Advanced Very High Resolution Radiometer - AVHRR, and the

Moderate-Resolution Imaging Spectroradiometer - MODIS) are used for detecting and

monitoring volcanic clouds (Holasek and Self, 1995; Woods et al., 1995; Prata, 2008; Clarisse et

al., 2012; Theys et al., 2013), but they cannot profile the atmosphere vertically and

measurements are affected by the presence of other types of clouds. Research aircraft are very

useful for getting information about the ash extent and concentration. They provide accurate

products, but they are not operational, the spatial coverage is limited, and technical limitations

are the same as for commercial aircraft, i.e., they cannot fly where the ash concentration is too



high. Ground-based instruments such as lidars (Sawamura et al., 2012), radars (Harris and Rose,
1983), and cameras are also important for monitoring the eruptions, but they are too sparse and
with limited spatial coverage.
Many techniques have been developed for detecting ash clouds (Prata, 2008; Clarisse et al.,
2012) and $SO_2$ clouds (Prata, 2008; Theys et al., 2013) relying on different satellite
measurements with different resolutions such as the Global Ozone Monitoring Experiment
(GOME-2), the Ozone Monitoring Instrument (OMI), the Infrared Atmospheric Sounding
Interferometer (IASI), MODIS, and the Atmospheric InfraRed Sounder (AIRS). The Cloud-
Aerosol Lidar with Orthogonal Polarization (CALIOP) on board of the Cloud-Aerosol Lidar and
Infrared Pathfinder Satellite Observations (CALIPSO) satellite is able to profile the volcanic ash
cloud with very high vertical resolution (Vernier et al., 2013), but the temporal resolution is not
adequate for following the development of the plume and sometimes the discrimination of ash
plumes from other type of clouds is problematic.
The GNSS RO technique is highly complementary to these other systems, enabling measurement
of atmospheric density and temperature structure in nearly any meteorological weather
conditions, during day and night, with global coverage, and with high vertical resolution and
high accuracy (e.g., *Anthes et al.,* 2011; Steiner et al., 2011). Several GNSS RO missions are
operating at present, providing vertical atmospheric profiles with good global coverage in space
and time, like the US/Taiwan FORMOSAT-3/COSMIC six-satellite constellation (Anthes et al.,
2008) or the European Meteorological Operational (MetOp) satellite series (Luntama et al.,

106 2008).

The use of RO data in numerical weather prediction has improved weather forecasting especially
in remote and data sparse areas of the globe (e.g., Cardinali, 2009) as well as tropical cyclone
track forecasting (e.g., Huang et al., 2005). Moreover, RO can deliver accurate information on
the thermal structure and cloud top altitude of convective systems and tropical cyclones as
demonstrated recently by Biondi et al. (2012; 2013; 2015). Monthly RO climatologies were
recently also used, together with radiosonde and reanalysis data, in a study aiming to detect
temperature effects of minor volcanic eruptions over 2001–2010 (Mehta et al., 2015). Due to its
characteristics, RO is a potentially valuable technique to study the structure of volcanic clouds
and to complement current monitoring systems. In this study we investigate whether the cloud



top detection technique developed by Biondi et al. (2013) can be applied as well for detecting
and monitoring volcanic clouds and for determining their cloud top height, their thermal
structure and influence on short-term climate.

**3.  Data Sets Used**
**3.1 GNSS Radio Occultation Data**
For this study we used RO temperature profiles processed by the Wegener Center for Climate
and Global Change (WEGC) with the Occultation Processing System (OPS) version 5.6
(Schwärz et al., 2013), based on excess phase and orbit data version 2010.2640 from the
University Corporation for Atmospheric Research (UCAR). The data have a vertical resolution
of about 100 m in the lower troposphere to about 1 km in the stratosphere (Gorbunov et al.,
2004). The quality of RO measurements is best in the Upper Troposphere and Lower
Stratosphere (UTLS) with an accuracy of 0.7 K to 1 K between 8 km and 25 km for individual
temperature profiles (Scherllin-Pirscher et al., 2011).
RO data from the following RO missions were used: CHAllenging Minisatellite Payload
(CHAMP) (Wickert et al., 2001), Satélite de Aplicaciones Científicas (SAC-C) (Hajj et al.,
2004), Gravity Recovery And Climate Experiment (GRACE-A) (Beyerle et al., 2005),
FORMOSAT-3/COSMIC, MetOP, and TerraSAR-X (Wickert et al., 2009). RO data from
different missions are highly consistent and agree within 0.2 K between 4 km and 35 km for
temperature (Scherllin-Pirscher et al., 2011), which allows merging of the data without any
calibration or homogenization (Foelsche et al., 2011; Steiner et al., 2011). Available RO data
products include individual profiles as well as gridded climatologies (e.g., Ho et al., 2012;
Steiner et al., 2013).
**3.2 AIRS and OMI Data**
We used ash observations from AIRS and $SO_2$ observations from OMI to identify volcanic
clouds and to differentiate between volcanic ash clouds and $SO_2$ clouds (see section 4.1). AIRS
is a thermal infrared (IR) sensor (Aumann et al., 2003) on-board the Aqua satellite, OMI is an
ultraviolet-visible (UV-Vis) spectrometer (Levelt et al., 2006) onboard Aura. Both polar orbiting
satellites operate in nadir mode (with footprints of 15 km in diameter and of 13 km x 24 km,



respectively). AIRS measures the spectrum of the thermal radiation emitted by the Earth-
atmosphere system (at wavelengths from 0.4 µm to 1.0 µm and from 3.7 µm to 15.4 µm, during
day and night). OMI measures the solar irradiance spectrum (i.e., light backscattered by the
atmosphere or reflected by the Earth during daytime) at wavelengths from 270 nm to 400 nm,
where $SO_2$ has strong and distinctive absorption bands. The OMI $SO_2$ retrieval (Yang et al.,
2007) provides integrated $SO_2$ concentrations expressed in Dobson Units (1 DU = 2.69 x $10^{16}$
molecules/cm²).
A selective detection of ash from AIRS is used in this study based on a robust volcanic ash
detection method (Clarisse et al., 2013) differentiating ash from clouds, sand and other dust. The
AIRS ash index detection has three levels of confidence (low, medium, high). A pixel with a
high level of confidence indicates that the presence of ash is almost certain. Note that the ash
concentration is not provided and that this very selective ash detection is not effective for low
ash concentrations. More details about ash and $SO_2$ products and their limitation are reported by
Brenot et al. (2014).
**3.3 CALIPSO Data**
We used level 1 total attenuated backscatter products from CALIOP (CAL_LID_L1, version
V3.01). CALIOP is a two wavelength (532 nm/ 1064 nm) lidar onboard the CALIPSO satellite
with a vertical resolution of 30 m/ 60 m and a horizontal resolution of 330 m/ 1000 m,
respectively, in the UTLS up to 20 km altitude (Winker et al., 2009). CALIOP attenuated
backscatter data were used for detecting the ash cloud altitude with high accuracy. The altitude
where the attenuated backscatter at 532 nm is, from top downward, starting to be larger than the
background noise is considered to be the cloud top altitude.
**3.4 MODIS Data**
MODIS is an imaging spectroradiometer flying aboard the Terra and Aqua spacecraft. The wide
spectral range of MODIS allows monitoring physical and optical cloud properties with global
coverage (King et al., 2013). We used NASA MODIS Atmosphere Images Hi-Res Global
Mosaic cloud data for defining clear air conditions and conditions with deep convection by using
the cloud top pressure (MYD06_L2 and MOD06_L2) as reference (http://modis-
atmos.gsfc.nasa.gov/index.html).




## 4. Study Cases and Methods

### 4.1 Volcanic Eruption Events

We have analyzed two eruptions with different characteristics as respective study cases: the Puyehue eruption in 2011 , which was mainly an ash eruption, and the Nabro eruption in 2011, which was mainly an $SO_2$ eruption.

Puyehue erupted on 5 June 2011 in Chile (40.35°S, 72.07°W). This eruption affected the Southern Hemisphere with its ash cloud spreading 360 degrees in longitude and finishing its first circle around the globe on 18 June 2011. Several flights in the Southern Hemisphere were cancelled due to the ash in the atmosphere.

During the night of 12 to 13 June 2011 an explosive eruption occurred at the Nabro volcano located in Eritrea (13.37°N, 41.70°E). This has been recognized as the largest stratospheric sulfur injection since Pinatubo (1991) (Bourassa et al., 2012; Robock, 2013), spreading mainly $SO_2$ in the atmosphere more than 60 degrees in latitude and more than 100 degrees in longitude within a few days and lasting for more than 15 days.

### 4.2 Methods

For the selected eruption cases we first located the ash and $SO_2$ clouds using the AIRS ash index (considering high level of confidence only) and OMI $SO_2$ data, respectively, as illustrated in Fig. 1 (left panels). In a second step, we screened all RO profiles at mean tangent point locations and selected those located within the region of the volcanic cloud as defined from AIRS and OMI data for each day after the eruption. Over a time period of 20 days from the eruption we found a total of 1109 profiles co-located with the Puyehue cloud, and 248 profiles co-located with the Nabro cloud, respectively.

For detecting the cloud top altitude and for analyzing the volcanic cloud structure we applied the anomaly technique developed by Biondi et al. (2013) for cloud top detection of convective (water) cloud systems and cyclones. We computed the bending angle anomaly by comparing each selected RO bending angle profile in the volcanic cloud area to the monthly RO reference climatology for the same area, i.e., subtracting the RO reference climatology profile from the individual profile and then normalizing with respect to the monthly reference climatology in



order to obtain a fractional (percentage) anomaly profile. The cloud top altitude is represented as
pronounced anomaly in the vertical bending angle structure.
The criterion chosen for cloud top detection is a bending angle anomaly variation larger than 3%
within a 2 km altitude range, in line with the experience from previous studies (Biondi et al.,
2013; 2015) and as found robust in sensitivity tests. We also computed the corresponding
temperature anomaly profiles in order to assess the impact of the volcanic cloud on the
atmospheric thermal structure. The reference climatologies for bending angle and temperature
were obtained by averaging all RO profiles collected in the period 2001 to 2012 to monthly
means, using a resolution (i.e., averaging cell size) of 5° x 5° in latitude and longitude, with
about 100 to 400 profiles averaged per grid cell (the specific number depending on month and
latitude). The climatology is provided at a vertical sampling grid of 100 meters sampled at 1° x
1° in latitude and longitude.
The cloud top altitude detected with RO was validated by using co-located CALIOP cloud top
data from attenuated backscatter within a spatial distance of 200 km. Although no CALIOP
measurements are available for the first days of the Puyehue and Nabro eruptions, we found
three RO-CALIOP co-locations for the Nabro cloud and seven RO-CALIOP co-locations for the
Puyehue cloud for the period 15–19 June 2011.

**5. Results and Discussion**
The results show that in case of both, ash and $SO_2$ volcanic clouds, the applied anomaly
technique works well. Figure 1 (top-right) presents volcanic cloud top altitudes detected from
RO observations for the investigated eruption cases of Puyehue and Nabro. The monthly
climatological tropopause in the respective regions is at 10.8 km and 17.1 km altitude,
respectively, as computed from RO data (Rieckh et al., 2014). The detection of cloud top
altitudes with RO is confirmed with highly accurate reference data from CALIOP observations
in Fig. 1 (bottom right). The comparison of cloud top altitudes from RO with co-located
CALIOP observations shows good agreement for Nabro and Puyehue with a correlation
coefficient of 0.94 and a root mean square (r.m.s.) error of 930 m. Though only 10 co-location
pairs were available for this comparison, the r.m.s. error is still quite favorable and fully
consistent with the findings for tropical cyclones and convective systems (Biondi et al., 2012,



2013) and reflects the co-location criterion of 200 km and the different vertical resolution of the
observation methods.
In Figure 2 we show the temperature and bending angle anomaly profiles before (left panels) and
after (right panels) the Puyehue and Nabro eruptions as examples of ash and $SO_2$ cloud effects,
respectively. The vertical structure of RO temperature anomaly profiles for the Puyehue eruption
(Fig. 2b) reveals a prominent cooling of about –2 K by the volcanic cloud at about 11 km in
agreement with the findings of previous studies with meteorological satellite data (Woods and
Self, 1992; Woods et al., 1995) and with a small number of RO data (Wang et al., 2009; Okazaki
and Heki, 2012). The cooling corresponds with a strong positive anomaly in bending angle (Fig.
2d). However, it is not possible to discriminate between volcanic ash clouds and convective
clouds from RO only, since the cloud top cooling is common for all convective processes
(Biondi et al., 2012, 2013, 2015). For the Puyehue eruption (ash cloud), we thus detected the
cloud top altitude, but we did not find any clear signature of the volcanic ash in the RO profile.
For discrimination of the clouds, additional information on ash is therefore needed, as used in
this study.
For the Nabro eruption the analysis was more complex because of the emission of significant
amounts of $SO_2$. Also the atmospheric structure was at the same time affected by the presence of
a low tropospheric aerosol cloud influencing the mid-tropospheric temperatures. The
tropospheric inversion feature near 6 km altitude in the Nabro case before and after the eruption
(Fig. 2 e,f) is a persistent feature from May to September and is due to dust clouds from pre-
monsoon dust storm activity (e.g., Posfai et al., 2012; Alharbi et al., 2013). We validated this
feature in the RO profiles with aerosol cloud top altitude information from CALIOP backscatter
data and found the cloud top altitudes consistent for all investigated months (see also Figs. 3 to

256     5).

RO temperature anomaly profiles (Fig. 2e) and bending angle anomaly profiles (Fig. 2g) just
before the eruption (1 June to 11 June 2011) in the area of Nabro (10° x 10° in latitude and
longitude) show a negative temperature anomaly of about 2.5 K at about 17 km, which occurs
close to the monthly climatological tropopause level (black dashed line).
During the Nabro eruption we detected the cloud top altitude and, furthermore, we also found a
clear signature distinguishing the eruption itself as shown in Fig. 2f for temperature and in



Fig. 2h for bending angle anomaly profiles co-located with the volcanic cloud (in a 5° x 5° box)
just after the eruption. A warm anomaly of nearly 4 K above the monthly climatological
tropopause appears as the eruption signature. The volcanic cloud tops (bending angle anomaly
peaks) correspond in this case to the primary tropopause (pink area) and the tropopause itself
corresponds to the secondary tropopause (cyan area). These results show that also in the case of
volcanic eruptions, as for tropical cyclones (Biondi et al., 2015) and convective systems (Biondi
et al., 2012), a double tropopause feature is found, where the lower level is caused by the cloud
top and the higher level represents the actual tropopause, which is pushed up by the strength of
the eruption.
The Nabro eruption cloud tops are located at a mean altitude of 16.3 km (Fig. 2f,h, violet dashed
line), which is below the climatological tropopause of 17.1 km (Fig. 2f,h, black dashed line). The
warming in the lower stratosphere appears just after the eruption, suggesting that the $SO_2$ cloud
directly reached the stratosphere as reported also by Fromm et al. (2013), Vernier et al. (2013)
and Fromm et al. (2014) and that its direct radiative effect induced a stratospheric heating. This
is different from the Puyehue case where there was no $SO_2$ but an ash cloud which induced a
cooling rather than a warming.
The question that arises is whether these thermal structures are really different and
distinguishable from normal atmospheric conditions. Figure 3 provides an overview on the
atmospheric structure under climatological conditions showing the monthly mean temperature
and bending angle anomalies for May 2007–2013 and June 2007–2013 for the areas of Puyehue
and Nabro. In the Puyehue region, monthly mean temperature anomalies are within about
± 1.5 K. In the Nabro region, the monthly mean temperature anomaly in the UTLS reaches
colder values in May (about –2 K ± 1.5 K) than in June (about 1 K ± 1.5 K) due to higher
convective activity.
In Fig. 4 we furthermore show the situation for the areas of Puyehue and Nabro in June 2010,
one year before the eruptions when no volcanic clouds were present. We analyzed the
meteorological conditions for both areas using MODIS data. We selected profiles in a deep-
convective environment (green) and in a non-deep-convective environment (blue) (denoting here
cloudy profiles with cloud top altitude lower than 300 hPa or clear sky). Figure 4 shows that the
June 2010 mean anomalies are similar to the climatological means. Temperature and bending



angle anomalies are larger in the presence of deep convective clouds while they are smaller in
the absence of deep convective clouds and do not differ that much from the climatology.
In the Nabro area it was very convective from 1 June to 11 June 2011 explaining why the
temperature profiles before the eruption show very cold anomalies (see Fig. 2e). Moreover, it is
shown that in the Nabro area the tropospheric inversion at about 6 km altitude is also present in
May and June monthly means (Fig. 3b) and in June 2010 under normal conditions (no volcanic
eruptions) (Fig. 4b).
Overall we find from Figs. 2 to 4 clear evidence that the mean anomaly profiles after volcanic
eruptions show a significantly different structure than those under climatological conditions.
There occurs a significant cooling of about 2 K in the mean after the Puyehue eruption (ash
cloud) and a significant warming of about 4 K in the mean after the Nabro eruption ($SO_2$ cloud).
The evolution of the atmospheric structure from May 2011 to December 2011 in the Nabro area
(Fig. 5) shows that the stratospheric warming in the area of the volcano remained for several
months. In May the average temperature anomaly in the UTLS was about –1 K. In June before
the eruption (green profiles) the average temperature anomaly reached about –2.5 K, but just
after the eruption (red profiles) the trend became opposite with a temperature anomaly of about
4 K in the mean, and of up to 10 K for individual profiles. The positive stratospheric temperature
anomaly in the Nabro area persisted until October 2011 and then decreased.
Nabro injected about 1.5 Mt $SO_2$ into the stratosphere that caused an enhancement of
stratospheric (hydrated sulfate) aerosol (Bourassa et al., 2012; Robock, 2013). Extended aerosol
layers up to 20 km altitude were measured for several months after the eruption, for the first few
weeks confined over North Africa and the monsoon region due to the monsoon anticyclonic
vortex and then spread over the larger Northern Hemisphere, causing warming of the lower
stratosphere (Bourassa et al., 2012). This aerosol enhancement likely explains the warming in the
Nabro region over a few months after the eruption as seen in Fig. 5.

**6. Conclusions**



Cloud structure and cloud top height are key parameters for the monitoring of volcanic cloud
movement and for characterizing eruptive processes and understanding the impact on short-term
climate variability.
We introduced a technique that uses as a first step observations in the thermal infrared (AIRS)
and UV-visible (OMI) for identifying volcanic ash and $SO_2$ clouds and for discriminating against
water clouds. In a second step we use observations from GNSS RO for detecting the cloud top
altitude and for analyzing the volcanic cloud structure. We demonstrated that the anomaly
technique developed by Biondi et al. (2012; 2013) for detecting cloud tops of convective systems
and tropical cyclones can also be used for detecting and monitoring volcanic cloud tops.
Volcanic ash clouds and $SO_2$ clouds have a different impact on the atmospheric thermal
structure. Our results revealed a cooling of about 2.5 K near the cloud top for ash clouds,
confirming previous findings. In contrast, we found a clear warming signature from $SO_2$ (and
hydrated sulfate) clouds after the eruption of Nabro, with mean amplitudes of about 4 K just after
the eruption and persisting for a few months.
From this encouraging evidence we conclude that, due to their independence from weather
conditions and due to their high vertical resolution, RO observations can valuably contribute to
improve detection and monitoring of volcanic clouds and to support warning systems. The high
accuracy and vertical resolution of RO observations for detecting the tropopause with global
coverage will also help to understand whether eruptions overshoot into the stratosphere and
contribute to short-term climate variability.
Several new RO missions are planned for the near future, like the COSMIC-2 constellation and
further RO receivers in the European MetOp and Chinese FY3 meteorological satellite series.
These, together with a much higher number of GNSS signals from the U.S. GPS, the Russian
Globalnaya navigatsionnaya sputnikovaya sistema (GLONASS), the European Galileo system,
and the Chinese Bei-Dou system will provide RO profiles with unprecedented coverage in space
and time for monitoring the thermal structure impacts of volcanic eruptions and their cloud
dispersions at any stage.

**Acknowledgements**





UCAR/CDAAC (Boulder, CO, USA) is thanked for providing access to its RO excess phase and
orbit data, ECMWF (Reading, UK) for access to its analysis and short-term forecast data. We
thank the WEGC processing team members for OPS development and for OPSv5.6 RO data. RO
data and the reference climatology used for this study are available at WEGC (via
www.wegcenter.at) and from the corresponding author (R.B.) on request. The research leading to
these results has received funding from the People Programme (Marie Curie Actions) of the
European Union's Seventh Framework Programme (FP7/2007-2013) under REA grant
agreement n° 328233. The authors thank L. Clarisse (ULB, Belgium) for providing ash
estimations.

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





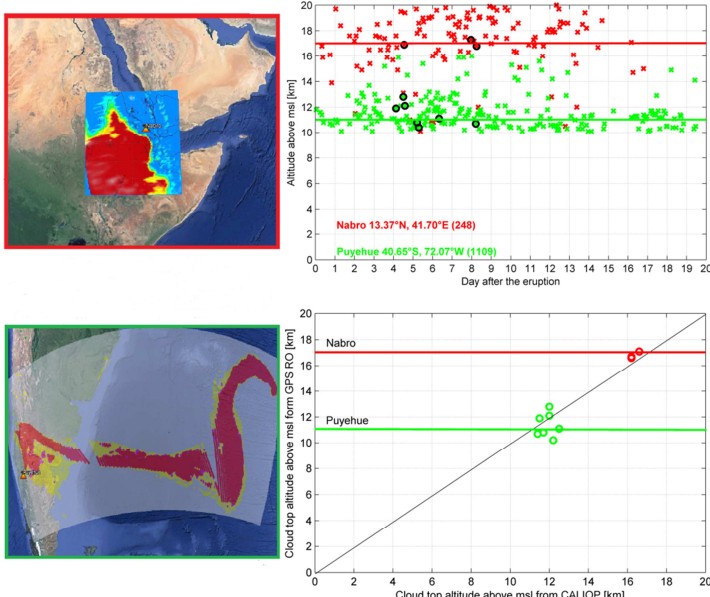


**Figure 1.** (top-left) SO$_2$ cloud from OMI data during the Nabro eruption, and (bottom-left) ash
index from AIRS data during the Puyehue eruption. (top-right) Cloud top altitudes of volcanic
plumes (cross symbols) for Puyehue (green), and Nabro (red), derived from RO data. (bottom-
right) Correlation between cloud top altitudes derived from RO with the closest cloud top
altitudes from CALIOP (circles). Horizontal solid lines denote the respective monthly
climatological tropopause altitudes for the three volcano locations. Numbers in brackets denote
the number of RO profiles.





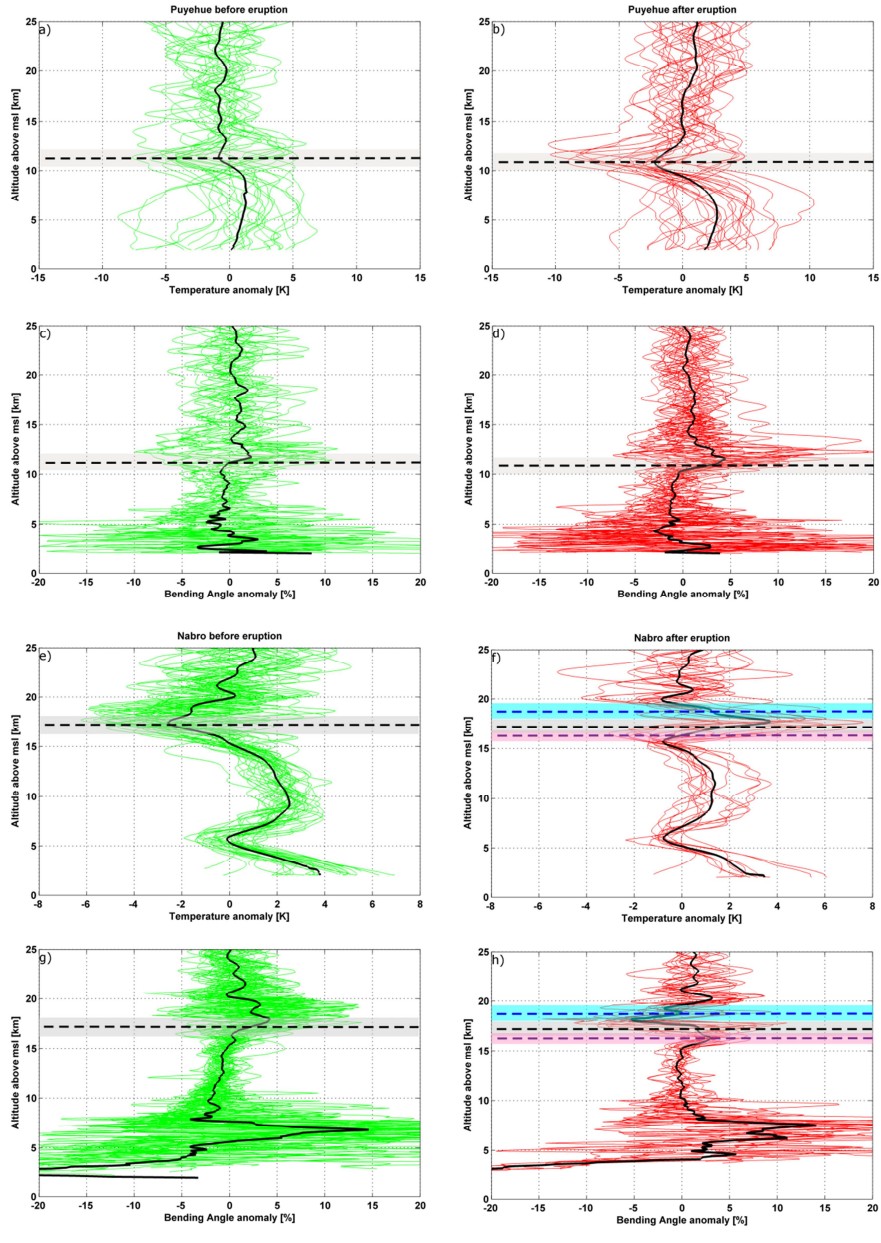


**Figure 2.** Puyehue (top four panels) and Nabro (bottom four panels) cases before (left column)
and after (right column) the respective eruption (Puyehue starting 5 June 2011, Nabro 12 June



2011). (a-b) Temperature anomaly profiles and (c-d) bending angle anomaly profiles in the area
of Puyehue before (green; May 2011) and after (red; 5-30 June 2011) the eruption, with the after-
eruption events co-located with the Puyehue eruptive cloud (AIRS ash index). (e-f) Temperature
anomaly profiles and (g-h) bending angle anomaly profiles in the area of Nabro before (green; 1–
11 June 2011) and after (red; 12–20 June 2011) the eruption, with the after-eruption events co-
located with the Nabro eruptive cloud (OMI $SO_2$). The mean anomaly profiles (black) and the
monthly mean climatological tropopause altitude (horizontal black-dashed lines), plus the
associated standard deviation of the individual-profile tropopause altitudes (shaded grey), are
also indicated. For the Nabro after-eruption events (f, h) in addition the mean primary tropopause
altitude (violet dashed line) and the mean secondary tropopause altitude (blue dashed line) are
shown, together with the corresponding standard deviations (pink and cyan shaded).









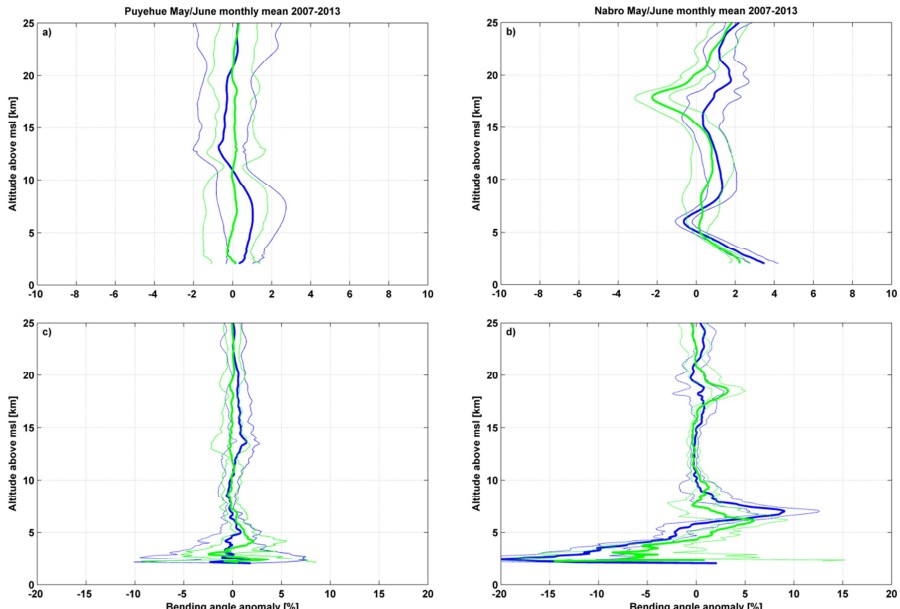


**Figure 3.** Monthly mean temperature anomaly profile (top panels) and bending angle anomaly

profile (bottom panels) averaged over May 2007–2013 (heavy green) and June 2007–2013

(heavy blue), and standard deviation of the individual monthly-means about this average for May

(light green) and June (light blue), in the area of Puyehue (a, c) and Nabro (b, d), respectively.

June 2011, the month of the eruption, is excluded.





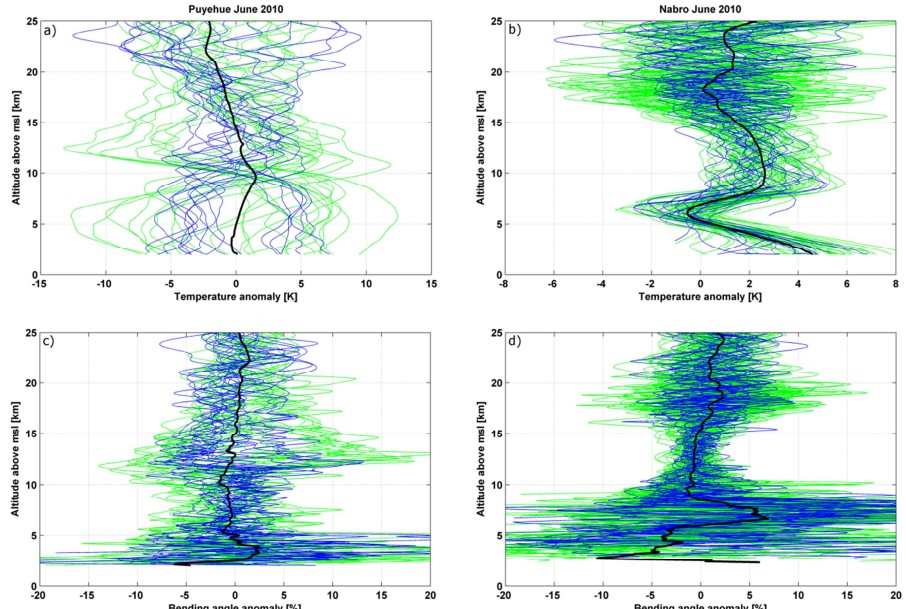

583

**Figure 4.** Individual temperature anomaly profiles (top panels) and bending angle anomaly profiles (bottom panels) in deep-convective environment (green), in non-deep-convective environment (blue), and mean anomaly profile for each profile ensemble (black), shown for June 2010 in the area of Puyehue (a, c) and Nabro (b, d), respectively.




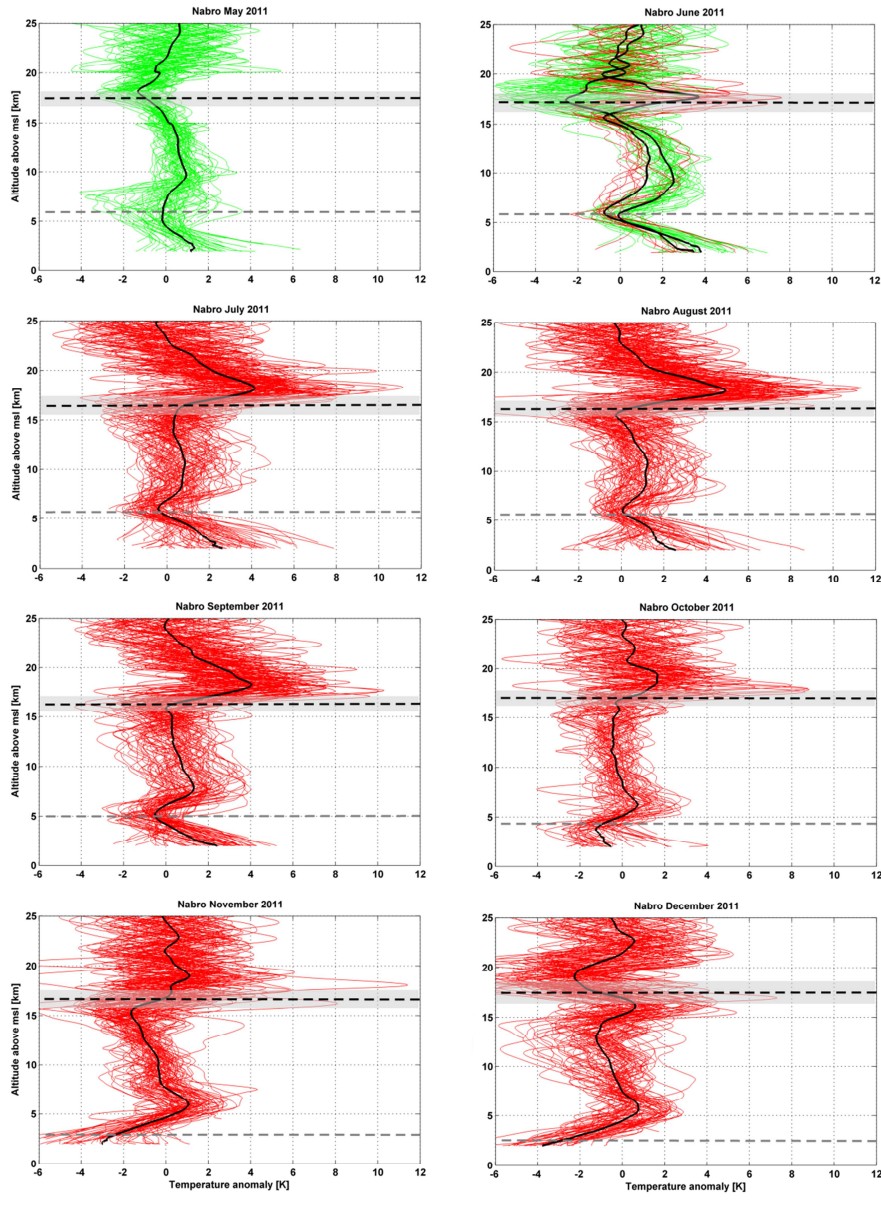


**Figure 5.** Individual temperature anomaly profiles before the eruption (green) and after the

eruption (red) with mean anomaly profile (black) in the area of the Nabro volcano (10 x 10



degrees box in latitude and longitude), showing the evolution of the thermal structure from
May 2011 to December 2011 (Nabro eruption in June 2011). Climatological tropopause altitude
for each month (black dashed line) with its standard deviation (shaded grey). The average
altitude of the tropospheric aerosol cloud from CALIOP measurements is indicated in each panel
(grey dashed line).