# Peer review of "A novel technique including GPS radio occultation for detecting and"

_Atmospheric Chemistry and Physics, 2015_

## Editor Comment (EC1) · S. A. Buehler (Editor) · 19 Jan 2016

I find the monthly mean anomaly figure (Figure 3) and its discussion in the text confusing. Is not the anomaly calculated relative to the monthly mean? Then the mean anomaly should be zero by definition. Please explain the concept behind this figure more carefully.

And a technical comment on the same figure: The top row misses x-axis labels.

---

## Referee Comment (RC1) · Anonymous Referee #3 · 1 Feb 2016

In this paper the author describe results on ash cloud detection as well as volcanic ash cloud top height determination using GPS radio occultation measurements of two volcanic eruptions (Puyehue and Nabro) in 2011. This is a well written and convincing study using two quite different eruptions, one being rich in ash and no $SO_2$ the other being very $SO_2$ rich. The paper however falls short in convincing me if this technique would also be applicable to eruptions including lower ash or $SO_2$ content. Admittedly, this was not the main aim of this work but the introduction builds on this argument (L36 and following), plus smaller eruption do threaten airways also considerably and techniques to monitor those are also necessary. The ability to detect smaller eruptions should in some way be addressed in the paper, the best place being most likely the

discussion section.

As mentioned by the authors the ash of the Puyehue eruption circled the earth leading to airspace closure quite far away from the eruption. It would be of particular interest up to what distance from the volcano RO techniques could be used to detect ash. This should also be addressed in the paper.

I have listed several references which should be included in this paper to give better cerdit to other work which has been done in this field.

Overall I feel this is an interesting paper and should be published after moderate to major revisions. Please find more specific comments below.

Specific comments:

L27: These references are ok, but there are some better ones for volcanic plumes reaching the stratosphere. A good reference could be the book by Sparks et al (1997) on Volcanic Plumes.

L31: The Pinatubo effect was as far as I know first published by MacCormick et al 1995, Nature, 373:399-404 and should be referred to in addition to the Robock paper.

L48: It is not the total ejected mass, but the mass flux which controls the height of the eruption cloud (see e.g. Woods, 1988, Bull. Volcanol, 50: 169-193). Furthermore eruption clouds typically overshoot the level of neutral buoyancy so there are certainly different height levels at which ash and aerosols are injected into the atmosphere during a single eruption.

L87: There is a quite comprehensive paper on observation of ash clouds using radar by Sawada 2004 (http://www.ofcm.noaa.gov/ICVAAS/Proceedings2004/pdf/entire-2ndICVAAS-Proceedings.pdf) that summarizes all observations of ash clouds with radar until 2004. This could be referenced here.

L99: From here on you refer only to RO techniques. Goals of your study are a) the
detection of volcanic clouds and b) the determination of cloud top height. L89-98 summarize briefly what has been done on the detection of ash clouds. Previous work on the determination of cloud top heights are missing however completely and there have been other approaches to determine cloud top height which should also be referenced here. Following is a list of papers which I feel should be included in your summary of the state of the art, as some techniques referred to in those papers (e.g. reflectance ratio measurements, photogrammetry) have not been referred to. (Chang, F.-L., et al., 2010. J. Geophys. Res. 115, D06208. doi:10.1029/2009JD012304; Dubuisson, P., et al., 2009. Remote Sens. Environ. 113, 1899–1911. doi:10.1016/j.rse.2009.04.018; Frey, R.A., et al., 1999. J. Geophys. Res. Atmospheres 104, 24547–24555. doi:10.1029/1999JD900796, Poulsen, C.A., et al., 2012. Atmos Meas Tech 5, 1889–1910. doi:10.5194/amt-5-1889-2012; Stohl, A., et al., 2011. ACP, 11, 4333-4351. doi:10.5194/acp-11-4333-2011)

L 214: I am not sure if I understand this correctly. For the reference climatology you average all profiles in an area of 5°x5°. Here you are referring to the climatology for the eruption in line 213 which is now sampled at 1°x1°. In case this is the eruption climatology than what is the possible error by substracting the average taken over 5°x5° which is a much larger area. But maybe I am misunderstanding this paragraph.

L215: Considering a spatial distance of 200 km between the CALIOP data and the volcano, can those profiles be considered representative for the cloud top, especially because the plume may have overshooted significantly near the vent. I note, for the main cloud at the neutral buoyancy level, though, this may be valid verification.

Fig1: Legends are missing on both maps. L555 there are no numbers in brackets. What are the black circles in the top right diagram?

Fig2c,d,e,g: From this figure it is very hard to see how often e.g. a certain bending angle has been measured in the individual profiles. Instead of plotting each single profile on top of each other, maybe a kind of density plot would be better in this case

indicating how often a certain bending angle has been observed at what height. This would also apply to Fig4bcd as well as to all panels in Fig5.

L230: It is worth to note that the RMSE of RO is comparable to the estimated ash cloud photogrammetry (see Genkova, I., et al, 2007. Remote Sens. Environ. 107, 211–222. doi:10.1016/j.rse.2006.07.021; Virtanen, T.H., et al., 2014. Atmos Meas Tech 7, 2437–2456. doi:10.5194/amt-7-2437-2014; Zakšek, K., et al., 2013. ACP. 13, 2589–2606. doi:10.5194/acp-13-2589-2013) Those should be referenced.

L239: The references to the work of Woods et al. are somewhat confusing. Woods and Self in their 1992 paper refer to studies by Maston as well as Harris et al. regarding the cooling effect, they only use this observation to state that this is also observed in their model. The way this paper is cited here one thinks Woods and Self did those observations which they did not. The same is actually true for the reference to the Woods et al, 1995 paper. Again this paper only used observations made by others, so again this reference should be removed and replaced by references to those papers where the processed the satellite data have been published first.

L 258: I am a bit confused, here it's a 10°x10° area, above it's a 1°x1° area (L214 and above).

L270: Instead of strength I would write buoyancy flux because the strength of an eruption is not well defined.

L317: Please provide a figure similar fig5 for the Puyehue eruption to see how this eruption evolved over time, especially because the eruption of Puyehue contained only volcanic ash.

L334: Is this method fast enough to be used as a real time monitoring system. Could it be used in a similar way as MODVOLC (doi:10.1016/j.jvolgeores.2003.12.008) from HIGP (http://modis.higp.hawaii.edu/) which is used to detect hotspots?

Fig4: In that figure you show one mean value (at least that how it looks like). Is that the

one for the deep or non-deep convective environment?

Fig5: Why are there 2 black lines in the upper right panel. I assume it is the average for the profiles before and after the eruption but this should be stated somewhere in the caption.
[Figure]

---

## Referee Comment (RC2) · Anonymous Referee #1 · 13 Feb 2016

This is an interesting and provocative paper that should be published after careful consideration of the reviewers' comments.

This reviewer is convinced that there is a volcanic signal in the RO anomaly profiles after the Puyehue (2011) and Nabro (2011) eruptions. The warming signature associated with the Nabro eruption, presumably due to the SO2 content of the plume, is especially convincing. The agreement of the RO estimated plume tops with the CALIOP measurements (Fig. 1) and the comparison of the anomaly profiles in the several months following the eruptions (Fig. 2) with the anomaly profiles in the same regions in the same months in years without eruptions (Fig. 3) support the claim that the RO anomaly profiles are providing evidence of the plumes.

While the results from two cases may not be 100% convincing, they are original and are compelling enough to be published in hope of stimulating further studies on other volcanoes.

My biggest concern is how soon after the eruptions do the horizontal scales of the plumes become large enough to produce atmospheric effects that are large enough to be detected by RO observations. With an RO observation horizontal averaging length scale (footprint) of 150 to 300 km, the scale of the plume would have to grow to at least 500-1000 km before RO observations would be likely to occur within the plume and detect its effects. The question is how long does it take for the plume to advect and disperse over a band 1000 km or greater in width (latitudinal extent)? A MODIS photograph taken 13 June, 2011, the day after the NABRO eruption, shows a plume spreading downwind towards the northwest with a width approximately 300 km 900 km downwind from the eruption.

**http://earthobservatory.nasa.gov/NaturalHazards/view.php?id=50988**

This suggests that the plume would be difficult to detect by RO the first day after the eruption, but would be detectable by RO within a few days after the eruption. A modeling study by Timmrect et al. (2003), as discussed in the modeling review paper by Textor et al. (2005) see Fig. 5, gives an indication of the rate of spread of the Pinatubo eruption (1991). By 9 days the plume is approximately 30° latitude wide (more than 3,000 km). By 16 days the plume covers much of the atmosphere between 30°S and 30°N, or more than 6,500 km wide. To the extent that these figures are representative, one would expect RO to be able to sample the plume starting a few days after the eruption, certainly after a week. This paper considers perturbations through 20 days after the eruptions, so I think there should be sufficient RO observations to detect the perturbations. The Pinatubo case is mentioned in the paper (Lines 186-188), but please clarify which volcano you are referring to that sentence "spreading SO2 in the atmosphere more than 60 degrees in latitude....". I think you are referring to Pinatubo not Nabro.
This sampling issue should be discussed by the authors. The review article by Textor et al. (2005) discusses the spreading of volcanic plumes and should be referenced (the current draft has no references to modeling studies.)

In the future, studies of this type would be greatly strengthened by model simulations of the volcanoes being studied. Of course this is beyond the scope of this initial paper.

Related to the sampling and scale issues are the photographs in Fig. 1. They are too small to be effective and there are no horizontal scales on them. I suggest replacing them with much larger, clearer photographs with a scale indicated (a  $5^{\circ}x5^{\circ}$  grid superimposed would be very useful). Some NASA photos of Puyehue are available at http://www.nasa.gov/topics/earth/features/20110606-volcano.html . The one at 15:37 UTC June 6 (the day after the eruption is a good one).

Related to the sampling issue, the authors should explain what is meant by "co-located" (Lines 194-194). For determining the co-located RO points, how are the Puyehue and Nabro clouds defined?

A few specific suggested edits:

Line 197 Add "with RO" after "....cloud structure"

Line 266 - I am not sure that "primary" is the best word to describe the first (lowest) tropopause, which is thought to be caused by the volcanic cloud, and "secondary" to describe the original or main tropopause. I would suggest something like "lower tropopause (pink area) and original (main) tropopause (cyan area)."

Line 281 – I suggest replacing "climatological" with "non-volcano" In this same paragraph, was the reference climatology from which the anomalies were computed the 2001-2012 mean (same as for the volcano anomalies)? If so, please say this.

Lines 335-336- I suggest rewording "RO observations can contribute to improved detection and monitoring..."

**ACPD**
The paper has very few typos-I found one in line 501 ("Determination").

References

Textor et al., 2005; numerical simulation of explosive volcanic eruptions from the conduit flow to global atmospheric scales. Annals of Geophysics, Vol. 48, Aug/Oct., p. 817-842.

Timmreck et al., 2003: Aerosol chemistry interactions after the Mt. Pinatubo eruption. In Volcanism and the Earth's Atmosphere, A. Robock, Ed., Am. Geophys. Un. Monograph Vol. 64, p. 155-177.

---

## Short Comment (SC1) · 15 Mar 2016

Comment on Biondi et al. (hereafter B16), **A novel technique including GPS radio occultation for detecting and monitoring volcanic clouds**.

As the title accurately states, this paper attempts to demonstrate that GPS-RO profiles may manifest a signal that suggests existence and height of volcanic cloud tops. B16 take advantage of their prior works that characterize organized (and horizontally large) convective cloud systems with GPS-RO data. Given that the volcanic eruption clouds in which they are presumably interested are also convective in nature, it is reasonable for them to hypothesize that a similar GPS-RO signal may be evident. If this effect is verified, the accurate, global, continuous, and fairly dense network of GPS-RO observations may well provide a new way of detecting and monitoring volcanic clouds.

My concern is that B16 has several fundamental weaknesses. Their current analysis is irreconcilable with the prior Biondi et al. papers' focus on active, opaque thunderstorm convective clouds. Moreover, B16 implicitly expand the definition of "cloud" beyond the active volcanic-convective eruption column to residual, optically thinner stratospheric sulfate and ash "clouds" that may last for weeks to months. This expansion of cloud scope comes without a consistent physical argument on how GPS-RO bending angle and temperature data would be impacted by this array of aerosol plumes and water-ice clouds.

The foundational papers by Biondi et al. make a very specific and limited claim; that widespread active convective water-ice cloud tops extending into the upper troposphere or overshooting into the stratosphere might be systematically accompanied by anomalously cold temperature. This cold anomaly owes to the fact that the top of the convective column represents the top of the moist adiabatic convective updraft with occasional excursion above the level of neutral buoyancy. Hence there could be a cold anomaly at the cloud-top altitude that is roughly proportional to the moist instability, vigor of the convection, and the horizontal scale (in order for it to be resolvable by GPS-RO measurements). These papers were careful to present independent cloud data with unassailable cloud top altitude resolution (e.g. CALIPSO) to prove the existence of the organized convection coincident with GPS-RO measurements. Their physical argument is that GPS-RO is neither directly nor deleteriously impacted by clouds or water vapor at UTLS altitudes; it is sensitive only to temperature and would show evidence of anomalously low temperature in relation to climatology at the cloud top. The stronger the anomaly (in T or bending angle), the more robust the evidence for water-ice cloud. The stronger the convective vigor and overshoot, the stronger the temperature anomaly. The inherent high vertical resolution of the GPS-RO data then is used to provide a precise estimate of the cloud-top altitude. Presumably in the absence of strong UTLS convective clouds the GPS-RO bending angle and temperature profile would not differ significantly from climatology, and the GPS-RO-determined tropopause height would coincide well with radiosondes and even gridded analyses. Moreover, even if there is vigorous, large-area, overshooting convection, **the effect would be local to the convective cloud cluster**. I.e. one would not expect the cold anomaly to persist in enough strength outside the cloud perimeter or after the cloud dissipates. This is my understanding of the principles argued in the Biondi et al. foundation papers. The prior papers did not venture into the subject of dust, sulfate, or ash-plume-top estimation with GPS-RO; neither does B16. They state in the abstract "an anomaly technique recently developed for detecting cloud tops of convective cloud systems can also be applied to the volcanic clouds." As they did in their prior papers, B16 need to show specific GPS-RO measurements collocated with and simultaneous to active volcanic-convective water-ice eruption clouds in order to be convincing. They did not make that case in this paper.

Part of Figure 2 is a detailed analysis near the Nabro location and eruption time. The pre- and post-Nabro panels of Figure 2 display GPS-RO profiles within a tight 5x5 lat/lon box centered on the volcano. The presumed intent of the tight cropping is an analysis similar to the prior Biondi et al. papers (and a presumed hypothesis consistent with their prior findings of a cooling at cloud top), focusing on the volcano-convective water-ice cloud. The pre- and post-Nabro GPS-RO signatures are essentially identical from the bottom to ~15 km, and above ~20 km. In between it appears that almost every post-Nabro profile is perturbed w.r.t. pre-Nabro. This is highly perplexing, for the following reasons. This small box does not represent the footprint of Nabro's continental-size plume footprint between 12-20 June, according to Clarisse et al. (2014) http://www.atmos-chem-phys.net/14/3095/2014/). Fromm et al. (2014) (http://onlinelibrary.wiley.com/doi/10.1002/2014JD021507/full) showed that the active Nabro eruption column top was near the tropopause only on 12-13 and 16 June and that the high-altitude eruption cloud in the volcano's vicinity was consistently situated northwest of the volcano (i.e. just a fraction of the box). Hence it is very perplexing to see that essentially all of the post-Nabro GPS-RO profiles in that box exhibit a UTLS anomaly with respect to pre-Nabro, and that all the perturbations are warming (rather than the cooling that a high convective water-ice cloud might give). Given the larger context of the injection-height chronology and the plume advection laid out in the above-mentioned papers, it would seem that the prescribed box and time interval would not contain such a stark, wholesale volcano-caused perturbation. If the post-Nabro assemblage of profiles in this box is correct, then I suspect that the anomaly must be attributable to non-volcanic forcing.

Regarding Puyehue, the ash map in Fig. 1 makes clear that the "cloud" that is being monitored reaches far beyond the range of the volcano's eruption column. Hence it is unconvincing that "an anomaly technique recently developed for detecting cloud tops of convective cloud systems" would have applicability. Moreover, B16 do not present a physical argument as to why volcanic ash would cool the surrounding atmosphere. As for Nabro, the Puyehue GPS-RO results are curious but unfounded on any physical argument. Hence the results here have insufficient substance to be convincing.

What I understand about the localized radiative impact of volcanic sulfates and ash is that they would **both** lead to **warming**, ash primarily through absorption of incoming solar radiation and sulfates via absorption of terrestrial IR radiation. Therefore one would expect both plume types to give the same sign of temperature impact. In either case the amount of GPS-RO-detectable warming would be strongly tied to the plume's physical size and concentration. In the absence of a fuller, alternate physical explanation for the cause-effect of these two plume types, the current explanations seem dubious. Moreover, making the case cannot be based on the assumptions and physical context embodied in the foundational Biondi et al. papers.

---

## Author Comment (AC1) · 21 Mar 2016

**Response to interactive editor comment by S. Buehler, doi:10.5194/acp-2015-974-EC1, 2016**

We thank the editor for his comments on Figure 3. Please find below a clarification and more detailed explanation of the concept behind this figure. We will improve the explanation in the revised paper accordingly.

**Comment:** *I find the monthly mean anomaly figure (Figure 3) and its discussion in the text confusing. Is not the anomaly calculated relative to the monthly mean? Then the mean anomaly should be zero by definition. Please explain the concept behind this figure more carefully.*

**Response:** We agree that the explanation of this Figure was not yet very good. In order to explain the concept behind the figure more carefully in the revised paper we will therefore update both the Figure 3 caption and title, and the related text in Section 5.

The caption will be improved to a more clear explanation like this: "Figure 3. Multiyear-averaged monthly mean temperature (top) and bending angle (bottom) anomaly profile, averaged over May 2007–2013 (heavy green) and June 2007–2013 without the month of eruption June 2011 (heavy blue), in the area of Puyehue (left) and Nabro (right). For indicating an estimated uncertainty range from interannual climate variability, also the standard deviation of the individual monthly mean anomaly profiles about the multiyear-average is depicted in all panels, as an envelope about the multiyear average profile, for both May (light green) and June (light blue)."

Along with this, the title over the panels will be improved to "Puyehue May 2007-2013 and June 2007-2013 w/o 2011" (over the left panels), and to "Nabro May 2007-2013 and June 2007-2013 w/o 2011" (over the right panels), respectively.

The related text in Section 5 will be improved like this: "The question that arises is whether these thermal structures are really different and distinguishable from normal atmospheric conditions. Inspecting for reference normal climatological background variability for May and June months without volcanic eruptions, Figure 3 provides an overview on the atmospheric anomaly structure under such climatological conditions. It shows the monthly mean temperature and bending angle anomalies averaged over May 2007–2013 and June 2007–2013 (excluding the month of eruption June 2011), and their estimated interannual variability during these years, for the areas of Puyehue and Nabro. In the Puyehue region, monthly mean temperature anomalies in the UTLS stay within about $\pm$ 1.5 K in both months. In the Nabro region, the monthly mean temperature anomaly above the tropical tropopause reaches colder values in May (about –2 K $\pm$ 1.5 K) than in June (about 1 K $\pm$ 1.5 K), due to higher convective activity."

These improved explanations should now make Figure 3 and its interpretation clear to the readers. Note that the construction of the temperature and bending angle anomaly profiles themselves (against a reference RO climatology 2001–2012) is explained in Section 4 before.

**Comment:** *And a technical comment on the same figure: The top row misses x-axis labels.*

**Response:** We will add the x-axis annotation to the top panels of Figure 3 again ("Temperature anomaly [K]"); they inadvertently disappeared in the final production of the discussion paper.

---

## Author Comment (AC2) · 21 Mar 2016

**Response to interactive short comment by M. Fromm, doi:10.5194/acp-2015-974-SC1, 2016**

The short comment is concerned that the present paper is not properly reconcilable with prior Biondi et al. papers, in terms of the anomaly approach used, and that the application of this anomaly technique now to "volcanic clouds" rather than "convective water-ice clouds" comes without adequate physical arguments as to the explanation of the anomaly signatures found.

Also there is an apparent misunderstanding of the RO profiles co-location approach used in the paper, which employs a core dataset of about 1300 RO profiles co-located with volcanic clouds and some complementary RO profile datasets from outside such clouds and outside eruption and post-eruption periods for reference purposes.

We thank the commentator for the effort of providing this input and think the following explanation of relevant key aspects of our approach—along with the associated improvements that we will include in the revised paper based on this short comment, an editor comment on Figure 3, and two referee comments—can clarify and alleviate the concerns:

The present paper is an exploratory feasibility study where we introduce a "novel technique *including* GPS radio occultation for detecting and monitoring volcanic clouds." *including* means, as already the abstract points out, that complementary ash and $SO_2$ data from radiometric measurements (AIRS, OMI) provide the location information to obtain the subset of those RO profiles only that are co-located with volcanic clouds. We can then use this core dataset (of about 1300 RO profiles) to learn about potential anomalous signatures from such clouds in the vertical thermal structure of the UTLS. This approach alleviates the concern that this co-located core ensemble may mix in RO profiles from outside such clouds. As a reference for these core results, several figures show the results in the context of complementary datasets taken intentionally from outside clouds and other time periods, in order to put the anomaly signature results from the core ensemble into context. For example, Figure 3 is of this sort, the explanation of which will also be further improved in the revised paper based on an editor comment.

Note also that we explain the characteristics of the input data and methods in Sections 3 and 4; we will also further improve there in the revised paper, related to making the definition of the geographic areas and collocation criteria for some of the datasets more clear.

*On concerns with the anomaly technique and physical explanation.* Technically we compute the anomaly profiles against collocated profiles extracted from a longterm reference RO climatology, i.e., we do it here the same way as for the Biondi et al. predecessor papers on convective cloud systems. We emphasize that this methodological approach (that we also briefly summarize again in the paper) is in no way dependent in its validity and viability on any physical arguments as to what causes some anomalous signature found—so that there is no basic concern for applying it, beyond its original Biondi et al use, also towards volcanic clouds. That is, irrespective of whether certain causative mechanisms induce warming (such as e.g. associated with absorption and local radiative heating by $SO_2$ clouds) or cooling (such as e.g. associated with the top of moist adiabatic convective updrafts), or any other anomalous signature, the anomaly technique will just help to isolate empirically whether there is some systematic deviation against normal climatological behavior for given datasets (e.g. cloud-effected vs outside-cloud).

Having found clear basic evidence for anomalous signatures associated with volcanic clouds, the physical interpretation is a separate matter, which we do in this initial paper intentionally only within a relatively modest tone and scope; more detailed future work will refine.

We finally note that we will improve some of the wording at a few places also related to this anomaly technique: in particular where the current wording "cloud top detection technique" is too sloppy we better refer to the technique more generally, i.e., beyond the specific aspect that anomaly profiles can be and are used to estimate cloud tops as one of the diagnostics.

---

## Author Comment (AC3) · 18 Apr 2016

Response to Anonymous Referee #1 (interactive comment received and published: 13 February 2016)

>>> *We thank the reviewer for the thorough review of our paper and for the constructive comments. Please find below our point by point response (in italics).*

This is an interesting and provocative paper that should be published after careful consideration of the reviewers' comments. This reviewer is convinced that there is a volcanic signal in the RO anomaly profiles after the Puyehue (2011) and Nabro (2011) eruptions. The warming signature associated with the Nabro eruption, presumably due to the SO2 content of the plume, is especially convincing. The agreement of the RO estimated plume tops with the CALIOP measurements (Fig. 1) and the comparison of the anomaly profiles in the several months following the eruptions (Fig. 2) with the anomaly profiles in the same regions in the same months in years without eruptions (Fig. 3) support the claim that the RO anomaly profiles are providing evidence of the plumes.
While the results from two cases may not be 100% convincing, they are original and are compelling enough to be published in hope of stimulating further studies on other volcanoes.

My biggest concern is how soon after the eruptions do the horizontal scales of the plumes become large enough to produce atmospheric effects that are large enough to be detected by RO observations. With an RO observation horizontal averaging length scale (footprint) of 150 to 300 km, the scale of the plume would have to grow to at least 500-1000 km before RO observations would be likely to occur within the plume and detect its effects. The question is how long does it take for the plume to advect and disperse over a band 1000 km or greater in width (latitudinal extent)? A MODIS photograph taken 13 June, 2011, the day after the NABRO eruption, shows a plume spreading downwind towards the northwest with a width approximately 300 km 900 km downwind from the eruption.
http://earthobservatory.nasa.gov/NaturalHazards/view.php?id=50988
This suggests that the plume would be difficult to detect by RO the first day after the eruption, but would be detectable by RO within a few days after the eruption. A modeling study by Timmrect et al. (2003), as discussed in the modeling review paper by Textor et al. (2005) see Fig. 5, gives an indication of the rate of spread of the Pinatubo eruption (1991). By 9 days the plume is approximately 30 latitude wide (more than 3,000 km). By 16 days the plume covers much of the atmosphere between 30 S and 30 N, or more than 6,500 km wide. To the extent that these figures are representative, one would expect RO to be able to sample the plume starting a few days after the eruption, certainly after a week. This paper considers perturbations through 20 days after the eruptions, so I think there should be sufficient RO observations to detect the perturbations.

>>> *We thank the reviewer for this valuable comment. We added further information on the RO measurement method, resolution, and RO sampling related to volcanic clouds. As discussed by the reviewer, the scale of 1000 km dispersion in 1 day is reasonable and confirmed by several works on different eruptions, e.g., Bignami et al.: Multisensor satellite monitoring of the 2011, Puyehue-Cordon Caulle eruption, Journal of Selected Topics in Applied Earth Observations and Remote Sensing, 7(7),*

*2786–2796, doi:10.1109/JSTARS.2014.2320638, 2014. Fig. 1 (upper left panel) of the manuscript shows that we found RO observations within the cloud from day 0 onward after the volcanic eruptions, based on the pre-defined cloud information from the radiometric data (OMI, AIRS). Whenever an RO observation is co-located with the cloud it will be possible to determine a signal of the cloud, with intrinsic horizontal averaging though. The larger the spread of the cloud and the longer its duration, the more RO observations will be co-located with the cloud.*

*We added the following sentence in section 4.1:*
*"Spreading about 1000 km in one day (Textor et al., 2005; Bignami et al., 2014), the extension of the volcanic cloud is well covering the scale of the GNSS RO horizontal resolution (about 200 to 300 km at the cloud altitudes) and atmospheric signals are large enough to be detected by RO observations."*

*In order to aid the understanding of relevant RO properties in a concise way, we included the following short description of the RO method and the resolution of RO observations in section 3.1, as follows:*
*"GNSS RO is an active limb sounding method using a satellite-to-satellite link. GNSS satellites transmit radio signals which are influenced by the Earth's refractivity field along their propagation path to a receiver on a Low Earth Orbit (LEO) satellite. Movement of the satellites leads to vertical scanning of the entire troposphere and stratosphere within about one minute and provides measurements with high vertical resolution but inherent along-ray horizontal averaging. The horizontal resolution across-ray is about 1.5 km and the along-ray resolution ranges from about 60 km in the lower troposphere to about 300 km in the stratosphere (Melbourne et al., 1994; Kursinski et al., 1997). The vertical resolution ranges from near 100 m in the lower troposphere to about 1 km in the stratosphere (Gorbunov et al., 2004). …*
*Data from the following RO missions were used: CHAllenging Minisatellite Payload (CHAMP) (Wickert et al., 2001), Satélite de Aplicaciones Científicas (SAC-C) (Hajj et al., 2004), Gravity Recovery And Climate Experiment (GRACE-A) (Beyerle et al., 2005), FORMOSAT-3/COSMIC (Anthes et al., 2008), altogether comprising about 2000 globally distributed RO profiling measurements per day."*

The Pinatubo case is mentioned in the paper (Lines 186-188), but please clarify which volcano you are referring to that sentence "spreading SO2 in the atmosphere more than 60 degrees in latitude. ". I think you are referring to Pinatubo not Nabro.

*>>> We are referring to the Nabro eruption here. To make this clear we improved the wording to:*
*"The Nabro eruption has been recognized as the largest stratospheric sulfur injection since Pinatubo in 1991 …, injecting mainly SO2 into the atmosphere, which spread…"*

This sampling issue should be discussed by the authors.

*>>> Please see the discussion above on the sampling issue; as explained there we have improved the text accordingly.*

The review article by Textor et al. (2005) discusses the spreading of volcanic plumes and should be referenced (the current draft has no references to modeling studies.)

*>>> We included this reference and cite Textor et al. (2005) in section 1 and section 2.*

In the future, studies of this type would be greatly strengthened by model simulations of the volcanoes being studied. Of course this is beyond the scope of this initial paper.

Related to the sampling and scale issues are the photographs in Fig. 1. They are too small to be effective and there are no horizontal scales on them. I suggest replacing them with much larger, clearer photographs with a scale indicated (a 5 x5 grid superimposed would be very useful). Some NASA photos of Puyehue are available at http://www.nasa.gov/topics/earth/features/20110606-volcano.html. The one at 15:37 UTC June 6 (the day after the eruption is a good one).

*>>> We thank the reviewer for this suggestion and for the provision of the links.*

*The purpose of Fig. 1. (left panels) is to show the reader an example of the radiometric measurements used in this study for supporting the co-location of RO profile observations with the volcanic plumes. We show $SO_2$ observations for Nabro and ash observations for Puyehue. After careful consideration we finally decided to keep these figures instead of showing images in visible channels, which are more easily available to readers anyway. However, we improved the two figures by enlarging them and by indicating latitude and longitude information.*

Related to the sampling issue, the authors should explain what is meant by "co-located" (Lines 194-194). For determining the co-located RO points, how are the Puyehue and Nabro clouds defined?

*>>> We reformulated the respective paragraph for better explanation and clarity. It now reads:*

*"For the selected eruption cases (Puyehue and Nabro) we used basic geographic areas of size 10° x 10° in latitude and longitude, centered at the volcano location. In a first step, we located the ash and SO2 clouds within these areas, using the AIRS ash index data for Puyehue (considering high level of confidence only) and the OMI SO2 data for Nabro, respectively, as illustrated in Fig. 1 (left panels). Spreading about 1000 km in one day (Textor et al., 2005; Bignami et al., 2014), the extension of the volcanic cloud is well covering the scale of the GNSS RO horizontal resolution (about 200 km to 300 km at the cloud altitudes) and atmospheric signals are large enough to be detected by RO observations.*

*In a second step, we screened all RO profiles at mean tangent point locations and selected those located within the region of the volcanic cloud as defined from AIRS and OMI data for each day after the eruption. Over a time period of 20 days from the eruption we found a total of 1109 profiles co-located with the Puyehue cloud, and 248 profiles co-located with the Nabro cloud, respectively. These cloud-collocated datasets were used as core RO datasets for exploring potential cloud-induced signatures. We complemented them by RO datasets from the same geographic areas outside clouds and during other time periods, in order to put the anomaly signature results from the cloud-collocated datasets into context."*

*For the reviewer's convenience we show below an example of a temperature profile and a bending angle profile (Figure 1r - left) located within the region of the Puyehue volcanic cloud as defined with AIRS ash observations (Figure 1r - right).*

[Figure]

*Figure 1r. (left) RO temperature anomaly profile (red) and bending angle anomaly profile (blue) co-located with (right) the Puyehue ash cloud on June 5, 2011. The ash cloud index is based on AIRS data with pixels of high confidence (red), medium confidence (magenta), and low confidence (yellow) (Clarisse et al., 2012). The green dot (right, within the cloud) indicates the location of the RO mean tangent point.*

A few specific suggested edits:

Line 197 Add "with RO" after ". . ..cloud structure"  *>>> Done.*

Line 266 - I am not sure that "primary" is the best word to describe the first (lowest) tropopause, which is thought to be caused by the volcanic cloud, and "secondary" to describe the original or main tropopause. I would suggest something like "lower tropopause (pink area) and original (main) tropopause (cyan area)."
*>>> This notation is usual one used for describing double tropopause features (e.g., Randel, W. J., D. J. Seidel, and L. L. Pan, 2007: Observational characteristics of double tropopauses. J. Geophys. Res., 112, D07309, doi:10.1029/2006JD007904). We therefore preferred to keep this notation.*

Line 281 – I suggest replacing "climatological" with "non-volcano" In this same paragraph, was the reference climatology from which the anomalies were computed the 2001-2012 mean (same as for the volcano anomalies)? If so, please say this.
*>>>We reformulated this paragraph for better clarity. It now reads:*
*"Inspecting for reference normal climatological background variability for May and June months without volcanic eruptions, Figure 3 provides an overview on the atmospheric anomaly structure under such climatological conditions. It shows the monthly mean temperature and bending angle anomalies averaged over May 2007–2013 and June 2007–2013 (excluding the month of eruption June 2011), and their estimated inter-annual variability during these years, for the areas of Puyehue and Nabro." …*

Lines 335-336- I suggest rewording "RO observations can contribute to improved detection and monitoring. . ."  *>>> Done.*

The paper has very few typos-I found one in line 501 ("Determination").   *>>> Done.*

---

## Author Comment (AC4) · 18 Apr 2016

Response to Anonymous Referee #3 (interactive comment received and published: 1 February 2016)

*>>> We thank the reviewer for the thorough review of our paper and for the helpful comments and reference suggestions. Please find below our point by point response (in italics).*

In this paper the author describe results on ash cloud detection as well as volcanic ash cloud top height determination using GPS radio occultation measurements of two volcanic eruptions (Puyehue and Nabro) in 2011. This is a well written and convincing study using two quite different eruptions, one being rich in ash and no SO2 the other being very SO2 rich. The paper however falls short in convincing me if this technique would also be applicable to eruptions including lower ash or SO2 content. Admittedly, this was not the main aim of this work but the introduction builds on this argument (L36 and following), plus smaller eruption do threaten airways also considerably and techniques to monitor those are also necessary. The ability to detect smaller eruptions should in some way be addressed in the paper, the best place being most likely the discussion section.

*>>>We performed some testing with smaller eruptions of Eyjafjöll and Etna, and found that the method works also for these cases. However, for these cases we did not find a statistically relevant number of GPS RO observations co-located with the volcanic clouds for more detailed analyses. This is the reason why we focused our feasibility study on the 2 largest eruptions occurring in the RO period since 2006. We added the following statement in section 6:*
*"We showed the feasibility for the two largest eruptions (Puyehue 2011, Nabro 2011) in the RO data period since the FORMSAT-3/COSMIC launch in 2006, where we analyzed about 1300 RO profiles for the two cases. We also found that the method works for smaller eruptions such as the Eyjafjöll eruption in Iceland in 2010 (Stohl et al., 2011). However, the number of cloud-collocated profiles was too small for a more detailed analysis, a situation that will drastically improve with the increased number of GNSS RO observations becoming available in future."*
*For the reviewer's convenience we show an example of cloud top results for Eyjafjöll in Fig.1r below. Evidently only 13 co-located RO profiles were available in this case, wherefore we did not include it as an extra example case; also CALIOP co-locations have not been available for this case.*

*As we briefly point out now more explicitly, this RO data coverage situation will drastically improve in the future with the increased number of RO constellation missions and other GNSS systems providing radio occultation signals in addition to the GPS. Motivated by this comment, we also included another brief paragraph now in the conclusions, pointing to the emerging Low Earth Orbit microwave and infrared-laser occultation methods, which are next-generation GNSS RO techniques that will enable a further drastic improvement of the characterization of volcanic clouds, as follows:*
*"Beyond the RO technique using the decimeter-wave GNSS signals, next-generation occultation techniques between Low Earth Orbit satellites are emerging that use centimeter- and millimeter-wave microwave signals (Kursinski et al., 2002; Kirchengast and Hoeg, 2004; Kursinski et al., 2009; Schweitzer et al., 2011a) and also micrometer-wave infrared laser signals (Kirchengast and Schweitzer, 2011; Schweitzer et al., 2011b; Proschek et al., 2011, 2014; Plach et al., 2015; Syndergaard and Kirchengast,*

*2016). These can simultaneously measure the thermodynamic structure, water and ice cloud structure, aerosol layering, and wind conditions and will therefore allow another drastic advancement of detecting and monitoring volcanic clouds by occultation methods in the future."*

[Figure]

*Figure 1r. Cloud top altitudes of volcanic plumes (cross symbols) for Nabro (red), Puyehue (green), and Eyjafjöll (blue) derived from RO data. Co-located CALIOP data are indicated (black circles). Numbers in brackets denote the number of RO profiles. Horizontal solid lines denote the respective monthly climatological tropopause altitudes for the three volcano locations.*

As mentioned by the authors the ash of the Puyehue eruption circled the earth leading to airspace closure quite far away from the eruption. It would be of particular interest up to what distance from the volcano RO techniques could be used to detect ash. This should also be addressed in the paper.

>>> *The presence of the cloud in general changes the atmospheric refractivity, which is the parameter influencing the GPS signal; this makes GPS RO observations useful and independent of the distance from the volcano. In this initial study we do not detect the presence of ash or $SO_2$ itself with RO; we rather use radiometric measurements first to obtain the cloud coverage knowledge. In a second step we then sample RO profiles co-located with the cloud and determine the cloud top altitude and the effect that different type of clouds have on the atmospheric thermal structure.*

*In order to aid the understanding of relevant RO properties in a concise way, we included the following short description of the RO method and the resolution of RO observations in section 3.1, as follows:*
*"GNSS RO is an active limb sounding method using a satellite-to-satellite link. GNSS satellites transmit radio signals which are influenced by the Earth's refractivity field along their propagation path to a receiver on a Low Earth Orbit (LEO) satellite. Movement of the satellites leads to vertical scanning of the entire troposphere and stratosphere within about one minute and provides measurements with high vertical resolution but inherent along-ray horizontal averaging. The horizontal resolution across-ray is about 1.5 km and the along-ray resolution ranges from about 60 km in the lower troposphere to about 300 km in the stratosphere (Melbourne et al., 1994; Kursinski et al., 1997). The vertical resolution ranges from near 100 m in the lower troposphere to about 1 km in the stratosphere (Gorbunov et al., 2004). ...*

*Data from the following RO missions were used …comprising about 2000 globally distributed RO profiling measurements per day."*

*And we added the following explanation on plume spreading and related sampling in section 4.1:*
*"Spreading about 1000 km in one day (Textor et al., 2005; Bignami et al., 2014), the extension of the volcanic cloud is well covering the scale of the GNSS RO horizontal resolution (about 200 to 300 km at the cloud altitudes) and atmospheric signals are large enough to be detected by RO observations."*

I have listed several references which should be included in this paper to give better credit to other work which has been done in this field. Overall I feel this is an interesting paper and should be published after moderate to major revisions. Please find more specific comments below.

Specific comments:

L27: These references are ok, but there are some better ones for volcanic plumes reaching the stratosphere. A good reference could be the book by Sparks et al (1997) on Volcanic Plumes.

*>>> We included this reference and cite Sparks et al. (1997) in section 1.*

L31: The Pinatubo effect was as far as I know first published by MacCormick et al 1995, Nature, 373:399-404 and should be referred to in addition to the Robock paper.

*>>> We included this reference and cite McCormick et al. (1995) in section 1.*

L48: It is not the total ejected mass, but the mass flux which controls the height of the eruption cloud (see e.g. Woods, 1988, Bull. Volcanol, 50: 169-193). Furthermore eruption clouds typically overshoot the level of neutral buoyancy so there are certainly different height levels at which ash and aerosols are injected into the atmosphere during a single eruption.

*>>> We corrected the sentence which now refers to Woods (1988) and reads:*
*"The mass flux of the eruption is fundamentally related to the maximum height reached by a volcanic plume (Woods, 1988)."*

L87: There is a quite comprehensive paper on observation of ash clouds using radar by Sawada 2004 (http://www.ofcm.noaa.gov/ICVAAS/Proceedings2004/pdf/entire-2ndICVAAS-Proceedings.pdf ) that summarizes all observations of ash clouds with radar until 2004. This could be referenced here.

*>>> Done.*

L99: From here on you refer only to RO techniques. Goals of your study are a) the detection of volcanic clouds and b) the determination of cloud top height.

*>>> Yes, the goals is as concisely summarized in the final sentence of this section (section 2).*

L89-98 summarize briefly what has been done on the detection of ash clouds. Previous work on the determination of cloud top heights are missing however completely and there have been other approaches to determine cloud top height which should also be referenced here. Following is a list of papers which I feel should be included in your summary of the state of the art, as some techniques referred to in those papers (e.g. reflectance ratio measurements, photogrammetry) have not been referred to. (Chang, F.-L., et al., 2010. J. Geophys. Res. 115, D06208. doi:10.1029/2009JD012304; Dubuisson, P., et al., 2009. Remote Sens. Environ. 113, 1899–1911. doi:10.1016/j.rse.2009.04.018; Frey, R.A., et al., 1999. J. Geophys. Res. Atmospheres 104, 24547–24555. doi:10.1029/1999JD900796, Poulsen, C.A., et al., 2012. Atmos Meas Tech 5, 1889– 1910. doi:10.5194/amt-5-1889-2012; Stohl, A., et al., 2011. ACP, 11, 4333-4351. doi:10.5194/acp-11-4333-2011)

>>> *We thank the reviewer for this information; we included also these references at this location in section 2 in our summary of the state of the art.*

L 214: I am not sure if I understand this correctly. For the reference climatology you average all profiles in an area of 5 x 5. Here you are referring to the climatology for the eruption in line 213 which is now sampled at 1 x 1 . In case this is the eruption climatology than what is the possible error by subtracting the average taken over 5 x5 which is a much larger area. But maybe I am misunderstanding this paragraph.

>>> *The reference climatology is provided at a 1° x 1° grid, based on averaging around each grid point over a cell size of 5° x 5° in latitude and longitude, to ensure a robust average and being representative for large-scale background field resolution.*
*We do not compute an "eruption climatology" but compute the difference of individual RO profiles (within volcanic clouds) minus the profile from the reference climatology, as described in section 4.2:*
*"We computed the bending angle anomaly by comparing each selected RO bending angle profile in the volcanic cloud area to the monthly RO reference climatology profile extracted for the same location, i.e., subtracting the RO reference climatology profile from the individual RO profile ..."*

L215: Considering a spatial distance of 200 km between the CALIOP data and the volcano, can those profiles be considered representative for the cloud top, especially because the plume may have overshooted significantly near the vent. I note, for the main cloud at the neutral buoyancy level, though, this may be valid verification.

>>> *This is right, but there is a typical cloud extension of several 100 kms for the main cloud so it is basically reasonable. And unfortunately we did not have any other more strictly co-located data option for independently validating our top estimation, since even with this 200 km criterion we could not get only a small number of co-locations. While we nevertheless consider the limited CALIOP results of Fig. 1 quite convincing in their consistency, this co-location issue of course leaves some "space-time representation uncertainty", which can hopefully be improved with more dense data in future.*

Fig1: Legends are missing on both maps.

*>>>We improved the two figure panels by enlarging them and by indicating latitude and longitude information.*

L555 there are no numbers in brackets. What are the black circles in the top right diagram?

*>>> We improved the caption of Fig.1 (top right panel). It now reads:*
*"Figure 1.…(top-right) Cloud top altitudes of volcanic plumes (cross symbols) for Puyehue (green) and Nabro (red), derived from RO data over the first 20 days from the eruption; co-located CALIOP data are indicated (black circles). Numbers in brackets denote the number of RO profiles. …"*

Fig2c,d,e,g: From this figure it is very hard to see how often e.g. a certain bending angle has been measured in the individual profiles. Instead of plotting each single profile on top of each other, maybe a kind of density plot would be better in this case indicating how often a certain bending angle has been observed at what height. This would also apply to Fig4bcd as well as to all panels in Fig5.

*>>> We thank the reviewer for this suggestion. We discussed and tested possibilities of alternative presentations, such as density plots. We then considered we will lose the height resolved amplitude information which is essential to show. We finally preferred to rather keep the present plot style. We inserted the information on the number of profiles before-eruption and after-eruption into the caption of Fig. 2, however.*

L230: It is worth to note that the RMSE of RO is comparable to the estimated ash cloud photogrammetry (see Genkova, I., et al, 2007. Remote Sens. Environ. 107, 211–222. doi:10.1016/j.rse.2006.07.021; Virtanen, T.H., et al., 2014. Atmos Meas Tech 7, 2437–2456. doi:10.5194/amt-7-2437-2014; Zakšek, K., et al., 2013. ACP. 13, 2589–2606. doi:10.5194/acp-13-2589-2013) Those should be referenced.

*>>> Done. We added the following sentence in section 5:*
*"These values also agree with estimates obtained with photogrammetry techniques (e.g., Genkova et al., 2007; Zakšek et al., 2013; Virtanen et al., 2014)."*

L239: The references to the work of Woods et al. are somewhat confusing. Woods and Self in their 1992 paper refer to studies by Maston as well as Harris et al. regarding the cooling effect, they only use this observation to state that this is also observed in their model. The way this paper is cited here one thinks Woods and Self did those observations which they did not. The same is actually true for the reference to the Woods et al, 1995 paper. Again this paper only used observations made by others, so again this reference should be removed and replaced by references to those papers where the processed the satellite data have been published first.

*>>> Done. We now cite Harris et al. (1981) and Matson (1984) instead.*

L 258: I am a bit confused, here it's a 10 x10 area, above it's a 1 x1 area (L214 and above).

>>> *Here we are specifically referring to the selection of the eruption cases where we investigate a 10° x 10° area around the volcano for volcanic clouds to sample a relevant number of individual RO profiles within the clouds.*
*We included an improved explanation at the beginning of the method section 4.2:*
*"For the selected eruption cases (Puyehue and Nabro) we used basic geographic areas of size 10° x 10° in latitude and longitude, centered at the volcano location."*

L270: Instead of strength I would write buoyancy flux because the strength of an eruption is not well defined.
>>> *Done.*

L317: Please provide a figure similar fig5 for the Puyehue eruption to see how this eruption evolved over time, especially because the eruption of Puyehue contained only volcanic ash.

>>> *We included this figure as new Figure 5 and included the following discussion in section 5:*
*"The further evolution of the atmospheric structure in the Puyehue area is presented in Fig. 5 until August 2011. It shows that the cooling after the volcanic eruption persisted for about two months into July 2011, whereas in August 2011 and subsequent months (no longer shown) the thermal structure had recovered and was back to normal climatological conditions (cf. Fig. 3a and Fig. 4a)."*

[Figure]

*Figure 5. Individual temperature anomaly profiles before the eruption (green) and after the eruption (red) with mean anomaly profile (black) in the area of the Puyehue volcano (10 x 10 degrees box in latitude and longitude), showing the evolution of the thermal structure from May 2011 to August 2011 (Puyehue eruption early June 2011). Climatological tropopause altitude for each month (black dashed line) with its standard deviation (shaded grey).*

L334: Is this method fast enough to be used as a real time monitoring system. Could it be used in a similar way as MODVOLC (doi:10.1016/j.jvolgeores.2003.12.008) from HIGP (http://modis.higp.hawaii.edu/) which is used to detect hotspots?

>>> *RO data are available in near real time, within about 3 hours, as needed and currently used for data assimilation in numerical weather prediction. In principle they could therefore be used in near real time to try getting earliest possible detection of volcanic cloud-induced anomalies in the atmospheric structure. At Wegener Center we intend to apply the detection and monitoring technique not in near-real-time but at a so-called fast track, on follow-on day of the observations. This still enables fast day-to-day tracking of emerging and evolving atmospheric volcanic effects, with one day latency but with climate-quality RO data processing with integrated uncertainty estimation and associated more robust diagnostics already (compared to near-real-time).*

Fig4: In that figure you show one mean value (at least that how it looks like). Is that the one for the deep or non-deep convective environment?

>>> *It is the mean anomaly profile for the whole dataset (convective + non convective). We corrected the caption of Fig.4 accordingly.*

Fig5: Why are there 2 black lines in the upper right panel. I assume it is the average for the profiles before and after the eruption but this should be stated somewhere in the caption.
>>> *We updated the caption (of Fig.6 in revised manuscript) accordingly: "Figure 6. … For June 2011 mean anomaly profiles are shown for pre-eruption and for post-eruption cases. …."*